# Understanding Barriers to Human Papillomavirus Vaccination among Parents of 9–10-Year-Old Adolescents: A Qualitative Analysis

**DOI:** 10.3390/vaccines12030245

**Published:** 2024-02-27

**Authors:** Denny Fe G. Agana-Norman, Monica Martinez Martinez, Manjushree Shanmugasundaram, Abbey B. Berenson

**Affiliations:** 1Department of Epidemiology, School of Public and Population Health, The University of Texas Medical Branch, 301 University Blvd., Galveston, TX 77555-0526, USA; 2Center for Interdisciplinary Research in Women’s Health, The University of Texas Medical Branch, Galveston, TX 77555-0587, USA; abberens@utmb.edu; 3School of Medicine, The University of Texas Medical Branch, Galveston, TX 77555, USAmashanmu@utmb.edu (M.S.); 4Department of Obstetrics and Gynecology, The University of Texas Medical Branch, Galveston, TX 77555, USA

**Keywords:** human papillomavirus, HPV, vaccine, vaccination, parental beliefs, vaccine hesitancy, school vaccination, early vaccination

## Abstract

HPV vaccination rates remain low among US adolescents, with only 54% completing the series in 2019. The vaccine is recommended at age 11–12 but can be given as early as age 9. Although it has been found that offering the vaccine earlier improves completion rates by age 13, parents remain reluctant to allow their younger children to initiate this vaccine. The purpose of this study was to better understand parental beliefs regarding receipt of the HPV vaccine among their children at ages 9–10. A 40 min phone interview was completed with 21 participants who were asked about their vaccine viewpoints. Even after receiving one-on-one education from a patient navigator, many caretakers expressed inadequate knowledge of the HPV vaccine and limited exposure to both positive and negative influences. The biggest concern was vaccine side effects, often resulting from a lack of medical understanding. Most parents were reluctant to vaccinate their children at a school-based clinic or pharmacy and believed that the government should not mandate HPV vaccination for public school attendance. Our study provides insight into parental beliefs and attitudes about HPV vaccination at age 9–10 years and barriers that need to be addressed.

## 1. Introduction

The human papillomavirus (HPV) is the most common sexually transmitted infection. About 13 million women and men in the United States (US) acquire HPV each year. HPV is a risk factor for genital warts as well as oropharyngeal, anal, penile, cervical, vulvar, and vaginal cancers [1,2]. Out of the 100 known HPV types, 13 are known to cause cancer. Genotypes for HPV 16 and 18 cause 70% of cervical cancers, while genotypes 6 and 11 cause genital warts [3,4,5].

High-risk HPV is responsible for approximately 2% of all cancers in men and 3% of all cancers in women in the US. HPV has been shown to cause about 90% of cervical and anal cancers, 70% of vulvar and vaginal cancers, and 60% of penile cancers [1]. Despite effective screening and the introduction of the HPV vaccine, over 13,000 women were diagnosed with cervical cancer in 2020 in the US, and nearly 4000 women died from this disease [5,6]. HPV-related cancers also have a significant economic burden; diagnosis and treatment of cervical changes and anogenital warts cost up to $2.9 billion a year [7].

It is estimated that HPV will infect 80% of men and women within their lifetime, usually as a result of sexual activity [1]. Thus, understanding the age at onset of sexual intercourse among US children and adolescents is essential to inform HPV vaccination recommendations. A 2011 NIH report found that about half of the US teenagers are sexually experienced, and a large group of teenagers report at least three lifetime sexual partners in the past year. Within 14-year-olds, the youngest cohort surveyed, 12.5% of females and 13.1% of males reported ever having sex [8]. However, various social and environmental factors correlate with the earlier onset of sexual intercourse. A meta-analysis in 2007 found that adolescents who have problems in school drink alcohol and face psychological complications: aggression for boys and depressive symptoms for girls were more likely to have sexual intercourse before the age of 15. Race, ethnicity, and gender play a role in the age of first intercourse. Black males are 2.5 to 3.5 more likely to have earlier initiation of first sexual intercourse compared to their adolescent peers; however, this pattern did not covary with school attitudes and educational aspirations. Sexual intercourse is more closely linked to religious attitudes and behavior among nonwhite girls and education level among white girls [9]. For this reason, it has been suggested that adolescents should complete all required doses of the HPV vaccine by age 13.

The Advisory Committee on Immunization Practices (ACIP) recommends starting HPV vaccination at 11–12 years of age, but it can be started as early as age nine. If not previously vaccinated, catch-up vaccination is recommended between 13 and 26 years of age [10]. Only two doses (6 to 12 months apart) are required if the individual is less than 15 years old when the first dose is administered. If the patient is older, three doses are recommended [10]. According to the Centers for Disease Control and Prevention, 53,000 cervical cancer cases could be prevented within the lifespan of girls younger than age 12 if HPV vaccination rates increased to 80% in eligible patients [11]. However, HPV vaccination rates continue to be suboptimal in the US even though the vaccine has been available for over 15 years. In 2019, 71.5% of adolescents 13–17 years old had received ≥1 dose of HPV vaccine, but only 54.2% had completed all required doses [11]. HPV vaccination up-to-date rates were even lower among males than females (52% versus 57%) in 2019. Asian and Hispanic adolescents, as well as adolescents with Medicare coverage, were more likely to be up to date in their HPV vaccination schedule than white adolescents, adolescents with private insurance, or those who were uninsured. Adolescent HPV vaccination rates also differ by state, from 31% of adolescents being HPV up to date in Mississippi to 79% in Rhode Island [12].

Factors associated with low vaccination rates include race/ethnicity, parental education, marital status, provider recommendation, income, and age [13]. Since the HPV vaccine may be initiated as young as age 9, parental beliefs significantly determine the age at which a child receives it. Parental HPV-vaccine decision-making is often influenced by concerns about vaccine safety, perceived low susceptibility to HPV infection, and lack of knowledge about HPV and the vaccine series [14]. A systematic review of studies from 1995 to 2007 of HPV-related beliefs and HPV vaccine acceptability studies showed that vaccine acceptability was higher when people believed the vaccine was effective, the vaccine was recommended by a physician, and when they perceived that HPV infection was likely to occur. Additionally, vaccine acceptability was higher in parents with lower educational levels. However, 6% to 12% of parents reported concerns that vaccination would promote adolescent sexual behavior [15]. A more recent study conducted in 2016 demonstrated that secondary acceptance of the HPV vaccine after an initial rejection was strongly associated with receiving follow-up counseling about HPV vaccination from healthcare providers, receiving a higher quality HPV vaccine recommendation during the initial discussion, and greater satisfaction with provider communication [16].

Vaccinating at earlier ages is medically beneficial in multiple ways. HPV vaccines had more than 99% efficacy when administered to women without prior exposure to HPV [17]. The HPV vaccine elicits a higher immune response when administered at ages 11 to 12 compared to later in adolescence [12]. When HPV vaccination is initiated at 9 to 10 years of age, there is a significant increase in up-to-date (UTD) status by age 13 [18]. Additional benefits include the consequential spacing of recommended vaccines and an increased emphasis on cancer prevention messaging [19].

In 2014, Dr. Abbey B. Berenson initiated a program in the pediatric clinics at the University of Texas Medical Branch to increase HPV vaccine uptake among children 9–17 years of age. Funding is provided by the Cancer Prevention and Research Institute of Texas to hire patient navigators (PNs) who educate families in person and offer the vaccine at no cost to those with an adolescent child receiving care at a participating clinic. This program has been highly successful in increasing both initiation and completion rates. However, we observed that parents of children 9–10 were less likely to initiate the vaccination series than those with 11–12-year-old children [20]. This study aimed to further examine parental attitudes towards earlier vaccine initiation by interviewing parents of 9–10-year-old children who received vaccination counseling through this program. Our secondary aim is to understand parental perspectives on general childhood vaccination and school-based vaccination initiatives to inform potential interventions and improve HPV vaccination rates in Texas.

## 2. Materials and Methods

With approval from the UTMB Institutional Review Board, all parents with children aged 9–10 at clinic sites were asked after receiving education on the HPV vaccine from a PN if they would be willing to participate in a qualitative interview. Flyers were additionally posted in clinics so parents or caretakers could ask if they were interested in the study. Parents who agreed to participate completed a form with their contact information. This was given to a research assistant who contacted them by phone or email to schedule an interview.

Data were collected through audio-recorded telephone interviews using semistructured interview guides between 3 October 2022 and 25 January 2023. All parents were informed that the interview was being recorded and asked to acknowledge their understanding. DF Agana trained three research assistants to conduct the phone-based interviews. Verbal consent was obtained from all parents prior to the interview. Interviews (n = 21) comprised 29 questions during a single telephone call and averaged 40 min each. A list of the IRB-approved questions has been provided in Appendix A. A $25 gift card was mailed to each participant as compensation for their time and effort. Each audio recording was transcribed and was reviewed by a second transcriber for accuracy. Formal data analyses were conducted between 6 February 2023 and 12 July 2023.

Independent evaluators (M Martinez Martinez and M Shanmugasundaram) created an initial coding scheme. The evaluators read the transcripts and written notes. Next, inductive codes were synthesized from emerging themes. Codes were then compared by an objective third-party validator (DFG Agana-Norman), who checked for accuracy and consistency. All analysis was conducted in Microsoft Word. A thematic analysis of vaccination beliefs was carried out based on the code summaries.

## 3. Results

Participants were recruited from the following UTMB clinic sites: Bay Colony, South Shore, Harborside, and Texas City. A total of 91 individuals were approached to participate, of which 41 agreed to be called for an interview, and only 21 caretakers completed the process. The participants’ children ranged from 9 to 10 years old (average = 9.8), with a 1:1 male-to-female ratio. Demographic characteristics of declined interviewees and completed interviewees are seen in Table 1. Approximately half of the parents interviewed identified as black, while the other half identified as white, with approximately one-fourth identified as Hispanic. Most declined interviewees identified as white and non-Hispanic.

### 3.1. General Vaccination Bias

All caregivers agreed that health was essential to their child’s life; however, two participants did not believe vaccines are generally safe for the public. A participant commented, “It depends on what it is, to me. I do not think all vaccines are good.” Yet, this belief did not translate directly toward their opinion regarding the advantages of HPV vaccination. All participants believed that HPV vaccination promoted their child’s health by avoiding disease, and only three participants’ children did not receive the HPV vaccine during their visit. One individual mentioned, “They [children] will not get sick with what comes with HPV later on because they will be vaccinated.” However, many parents (16 of 21) voiced concerns about possible side effects from vaccines, and thus, some parents tended to avoid certain vaccinations (Table 2). Specific vaccines mentioned were those for influenza and COVID-19.

### 3.2. Knowledge Gap

Half of the caregivers (10 out of 21) voiced concerns about possible vaccination side effects, demonstrating inadequate medical knowledge and processes surrounding medical innovations. One parent refused to have her daughters vaccinated against HPV due to an adverse reaction she had experienced. The parent stated, “It affected my ovaries…every month when I ovulate, I can tell which one is ovulating, and for about a week every month, my ovary feels like it’s on fire.” (Table 3). However, more than half of the caretakers expressed their lack of HPV information (12 out of 21) or exposure to outside positive or negative HPV resources (14 out of 21), mentioning, “I honestly have not heard anything at all.” (Table 3).

### 3.3. Patient Trust

Fourteen out of twenty-one caretakers voiced a strong distrust of news/social media for vaccination information. Parents rated these as the worst sources of vaccine information. Concurrently, individuals expressed a lack of trust in the medical system. It was found that 6 out of 21 individuals did not trust the healthcare system due to the short development time for the COVID-19 vaccine, mentioning, “I want to hear that we took our time, and we were able to pinpoint exactly what we needed and exactly what they needed.” (Table 4). Eight out of twenty-one parents explored the possibility of risky situations when undergoing medical processes: “I understood the risk when I got the COVID vaccine.” (Table 3). Lastly, 5 out of 21 individuals explained that their mistrust originated from the experimentation of people to understand treatment. A parent’s niece convinced him to receive the COVID-19 vaccine by mentioning, “Dude, hello, we all have to be guinea pigs sometimes.” (Table 4).

### 3.4. Physician Reliance

In contrast, over half of the caretakers expressed trust in doctors for information (18 out of 21). One participant even mentioned, “Anything the doctor says is good for my children, I’m all ears for.” (Table 5). Three out of twenty-one caretakers reported feeling more comfortable when medical professionals incorporate patients into their medical decisions: “That is the best way, kind of sharing the information.” (Table 5). Additionally, caretakers were hesitant to allow their children to be vaccinated at alternative locations without direct supervision of a physician (11 out of 21), stating, “I know how bad a reaction can be. I would rather it happen at the doctor.” (Table 5). Some parents (7 out of 21) preferred vaccine administration by the child’s general doctor rather than a specialty clinic. One said, “I think that really should be done by the main head doctor because, again, he is the one that has been seeing the same kid for however many years they know that they know the child”.

### 3.5. Attitudes about Alternate Vaccination Sites

Most parents expressed reluctance to have their children vaccinated at a school. Common explanations for this preference were “I don’t trust the cleanliness of schools,” “I just want them to have it where they’re the most comfortable and will behave the best while they’re getting it,” or “school is not a safe place in a sense for medical stuff.” Less than half of the participants (8 out of 21) would be comfortable with a nurse administering vaccines without a supervising physician, stating, “A nurse is also a trained professional.” (Table 6). However, few participants (3 out of 21) would consider school vaccination if this became a common practice. “It would make me more accustomed to it, I guess, like a cultural thing.” For similar reasons, many parents (9 out of 21) were also unwilling to vaccinate their children at a pharmacy; however, the rationale for this preference was vague. Explanations given were “No… I just don’t approve of it.” (Table 6). Some of those individuals who would not prefer to vaccinate their children at a pharmacy would do so if necessary (5 out of 21), explaining, “It’s not something I would not do, but I’d just prefer not to.” (Table 6). Furthermore, all the participants disagreed with the dentist administering the HPV vaccine, stating that it is unrelated to oral health.

Lastly, regardless of whether the participants’ child had received the HPV vaccine, most parents (17 out of 21) believed that the government should not enforce HPV vaccination as a prerequisite for public benefits. One of the participants mentioned, “Some people are religious, and they do not believe in vaccines, and by y’all doing that, that’s stopping another way from not attending school for people so that don’t need it, so that stops education for little ones in making the world dumb, honestly.” (Table 6).

When asked about school vaccination logistics, half of the parents stated they preferred to be informed of their children’s vaccination options at their school through the welcome package given out at the beginning of the school year, while others preferred an electronic notification. Participants were also asked how far in advance they should be notified about the availability of school vaccinations; the most frequent response (8 out of 21) reported was at least 2 weeks in advance, ranging from 1 week to 1 month.

### 3.6. HPV Vaccination Administration

Less than half of the individuals interviewed (8 out of 21) mentioned a preference for starting HPV vaccination during the teenage years (age above 10 years). The rationale varied widely between individuals. Some parents stated that HPV vaccination should start as soon as menstruation begins: “My daughter started early, so that’s why I was okay with her getting the HPV vaccine”. Other caretakers believed that HPV vaccination would encourage the onset of intercourse (2 out of 21) and thus should be administered when a child can talk about sexuality: “When you are able to talk to them about sex, and you’re able to feel comfortable talking about things like that.” (Table 7). Lastly, a couple of participants (5 out of 21) believed that the best time to administer the HPV vaccine is when one is sexually active: “I would probably recommend it to people who are active, sexually active, and then exploring their sexuality.” (Table 7).

## 4. Discussion

Our study found that caretakers’ opinions on the timing of HPV vaccination differed from the ACIP guidelines. Many participants preferred starting the HPV series during their child’s teenage years (>10 years). The rationale for this preference varied widely between individuals and was not necessarily scientifically supported. Many of the justifications given for delayed vaccination had to do with the misguided association between the HPV vaccine and sexual maturity. Much work has been carried out to deconstruct this narrative, including the rebranding of the HPV vaccine by the CDC-Hager Sharp campaign in 2015 [21]. Providers’ support and counseling continue to be key aspects in educating parents on this subject. When parents believe that a child’s vaccination could initiate sexual behavior, rather than disproving their argument, providers should attempt to desexualize the HPV vaccine by highlighting its use against cervical cancer and emphasizing its novelty [22]. When questioned on the necessity of its early administration, providers can emphasize the need to vaccinate before the onset of potential HPV infection. Another strategy is to inform caretakers that HPV vaccination provides a more robust immune response when administered at a young age. Additional assistance for physicians in navigating difficult discussions with parents can be found in the CDC HPV vaccination recommendations [23]. It should also be noted that most participants in our study agreed with starting early HPV vaccination. However, the interviews reported parents’ opinions after PN education had already occurred. It is, therefore, difficult to assess whether participants felt this way before the PN HPV vaccination educational session.

Overall, interviews of parents with 9- to 10-year-old children revealed considerable concern about vaccination side effects. This may have led parents to avoid certain vaccines, such as those to prevent influenza and COVID-19. Despite concern about vaccine safety, in general, most of the parents interviewed believed that the HPV vaccine was beneficial to their child’s health by protecting them against future diseases. These results conflict with the findings in the literature, which mention an overall increase in HPV vaccine hesitancy. For example, a questionnaire administered to 5249 women in Greece found a decline in intention to vaccinate when the HPV vaccine became available, coinciding with a rise in the proportion of women concerned about its potential side effects, specifically those promoted by the media [24]. It should be noted that while the participants in our study voiced a strong distrust in news/social media when obtaining vaccination information, side-effect hesitancy continues to be a strong influence against HPV vaccination. Since parents were interviewed after their child’s clinic visit, their responses likely reflect the education on HPV and the vaccine they received from a PN. In fact, 18 out of the 21 parents agreed to have their child receive the HPV vaccine the day of the visit. Therefore, our study suggests that educating parents on HPV and the HPV vaccine is an effective strategy to help overcome barriers to vaccination.

Throughout interviews, multiple caregivers demonstrated inadequate medical information or a lack of understanding of the medical process and vaccine functionality. School-based health education has consistently improved student health knowledge surrounding the scientific and philosophical principles of individual and societal health [25]. However, as of 2018, only 38 states and the District of Columbia mandate sex education and HIV education, and when these are provided, only 17 states require the content to be medically accurate. In Texas, where this study occurred, sex or HIV information is not mandated. If provided, it does not need to be medically accurate, only age-appropriate, and can allow for the promotion of religion [26]. None of the states consider vaccine literacy as an objective in their health education curriculum. Thirty-two states have no formal vaccine education, and few require it in their school health curriculum. Only Colorado and Maryland require health classes to discuss vaccination during middle and high school. Texas requires vaccine education during high school and not in middle school. Furthermore, none of the states provide vaccine guidance to teachers, which explains why only 37% discuss herd immunity in their classrooms [27]. These circumstances provide the platform for vaccine misinformation and vaccine illiteracy. Therefore, states need to change school health education content requirements before it could be another avenue to educate individuals on HPV and the vaccine.

Knowledge is critical in the beginning stages of behavior change, especially in adopting healthy behavior [28]. Key stakeholders (such as parents, school staff, and providers) should possess adequate knowledge about HPV and the vaccine to make correct decisions regarding their children’s or students’ health. Even though all participants had received education from a PN, they still reported a lack of HPV vaccine information and a lack of exposure to positive or negative sources of information. These results indicate that participants had not likely received HPV education before counseling by the PN. Apart from the physician’s office, a large portion of education is expected to be completed in schools. However, in 2017, the US Department of Health and Human Services released a report stating that only 1.2% of secondary schools in Texas provide the HPV vaccine, with a national range of 0–29%. Regarding general sexual health services, only 0.3% of Texas schools would provide STD treatment (0–19% range), and only 0.3% would provide condoms (0–20% range). Seventy-eight percent of Texas secondary schools did not provide sexual or reproductive health, and only 29% provided parents and families with information on how to communicate with their children about sex [29]. Vaccination education should be a multidisciplinary approach, as vaccination initiatives would likely suffer if public schools were not used as a communication channel.

Interview responses indicate that many participants mistrust medical advances, approaches, and vaccine testing. Some mistrust is rooted in the history of the medical profession, specifically regarding research and the use of people as subjects of experimentation. The infamous Tuskegee Syphilis Study between 1932 and 1972 left racial and ethnic minorities with generational trauma, continuously propagated by a health system that creates, sustains, and reinforces racism, classism, homophobia, transphobia, and stigma [30]. Half of the participants identified as black, and approximately one-fourth identified as Hispanic; data, however, was not stratified by race or ethnicity, making it difficult to know whether medical mistrust is related to minority status within this study. However, it should be noted that participants’ mistrust revolved around the theme of medical experimentation. A few participants believed that the longer it takes to produce a vaccine, the safer the vaccine. The rapid development of the COVID-19 vaccine fueled such misconceptions [31]. Other participants felt that medical processes and procedures were dangerous. Lastly, caretakers also felt that the medical industry uses people as a means to understand treatments. When interviewing a caretaker, he stated that his niece convinced him to take the COVID-19 vaccine by focusing on the overall public benefit and coming to terms with being considered a guinea pig. The participant’s feelings appeared to arise from the belief that the healthcare system in place does not prioritize an individual’s well-being or health. Individuals are, therefore, forced to see for their well-being and protect themselves from medical management. Medical mistrust can have serious consequences, such as lower healthcare utilization and poorer management of health conditions. It is the responsibility of the healthcare community to amend these wrongdoings by acknowledging past and ongoing trauma and collaborating among communities that have experienced historical and social exclusion [32].

Provider endorsement is an important determinant factor in HPV vaccine acceptance [33]. Physicians’ recommendations can increase HPV vaccination and completion by three and nine times [34]. These findings are consistent with our data, as caretakers thought doctors were a trustworthy source of medical information and vaccine administration. Moreover, caretakers preferred that their child’s physician supervise vaccination. These results indicate that parents prefer that their child be vaccinated in a clinical setting. Despite public trust in healthcare professionals, many providers lack general HPV and vaccine knowledge and are hesitant to make recommendations not stated in professional organizations’ guidelines. HPV vaccination initiatives should encourage doctors to internalize HPV vaccination guidelines from their colleagues and profession. Additionally, many providers believe they will not convince parents and patients of the vaccine’s effectiveness and value. An initiative focused on self-efficacy might address this issue by improving physician confidence in addressing parental concerns [34]. How vaccination is discussed can also significantly influence a parent’s decision to vaccinate their children. In a 2013 study, the American Academy of Pediatrics found that parental odds with vaccination increased when providers discussed vaccines with parents compared to when providers treated vaccines as routine [35]. Many participants reported feeling more comfortable when medical professionals incorporate patients into decisions regarding their care. However, a collaborative discussion, as well as shared decision-making, surrounding vaccinations is not typically executed by providers since doing so would present vaccine avoidance as a valid option. Yet, disregarding parental emotions in the matter could hinder the physician–patient relationship and, thus, future scheduled vaccinations. Therefore, a physician should tread carefully in this matter as a choice should be given to the patient or the patient’s parents while concurrently voicing a strong recommendation for the HPV vaccine.

Patient navigators can assist in creating a discussion with the patient or family in situations when providers cannot do so. PNs can take advantage of regular clinic visits to inform unvaccinated or partially vaccinated patients about HPV and the availability of a vaccine that protects against infection and disease. Such a program was initiated in 2014 at the University of Texas Medical Branch. The PNs hired by the vaccination program screened the electronic medical records of 9–17-year-old patients with upcoming appointments at a UTMB clinic and approached parents whose children were unvaccinated or incompletely vaccinated against HPV. If the parents consented, the children received the vaccine that day. The vaccine is administered at no cost to the family. The program’s high initiation and completion rates show that taking advantage of a routine clinic visit to educate patients on HPV and offer the vaccine is an effective strategy to increase vaccination rates [20].

Moreover, school-based vaccination is considered one of the most effective and efficient means of ensuring high HPV vaccination rates among adolescents [36,37]. Under this initiative, schools would routinely deliver recommended vaccine doses to students, making these prime locations for vaccine initiatives. HPV vaccination has historically been offered exclusively in clinics, which may create a barrier for caregivers. Offering vaccines in school settings at no cost has increased adolescent immunization rates in some areas of the US. However, only Rhode Island, the District of Columbia, and Virginia schools require individuals to receive the HPV vaccine. “Vaccinate before you Graduate” is a Rhode Island program where the Department of Health offers free vaccines to all middle and high school students. Rhode Island is now known to have the highest HPV vaccination rate in the country [36]. Replicating such a program elsewhere may be feasible as the Affordable Care Act requires insurance companies to cover vaccines recommended by the ACIP without copays. Furthermore, Vaccines for Children (VFC) is another federally funded program that can provide free vaccination to uninsured children or those not covered by Medicaid [38]. Despite greater accessibility, caretakers were reluctant to vaccinate their children at a school. The school’s cleanliness was questioned, as well as the qualifications of the school nurse and concerns surrounding the safety of the children. This is consistent with previous findings from a school influenza study [39]. For similar reasons, many caretakers were unwilling to vaccinate their children at pharmacies.

Some caretakers would consider school vaccination if this became a common practice, indicating that culture and social norms are potential barriers. These findings are like those seen in a study of parents of 11- to 14-year-old children attending seven middle schools in a large urban school district. Parents who had previously used a school vaccination program were 2.5 to 4 times more likely to report willingness to receive each specific vaccine at school than those whose children received immunizations at another site [40]. Therefore, various trials of school vaccination programs will need to occur before seeing a rise in caretaker willingness to change to this venue for vaccinations. The government plays a crucial role in implementing school vaccination programs. School vaccination programs in Canada, Australia, and some parts of the UK are funded by the government and have seen a significant increase in adolescent vaccination rates. However, the US does not currently have a culture supporting this agenda [41].

When considering governmental policy surrounding school vaccination, compulsory vaccination should be mentioned. Most participants believed that the government should not enforce HPV vaccination as a prerequisite for public benefits. Regardless of increased vaccine accessibility for adolescents and children, caretakers continue to have the right to refuse any vaccination that is not mandatory under state law. However, exemptions are made for individuals with medical conditions or based on religious freedom [41]. A 2019 systematic review of public opinions surrounding compulsory vaccination found that parents were more likely to refuse the HPV vaccine than any other adolescent vaccine. It was also found that support towards mandatory vaccination increased after their implementation, indicating a possible cultural shift that needs to be addressed [42].

## 5. Conclusions

The current ACIP recommendations state that the HPV vaccine should be given between 11 and 12 years of age. However, it can be initiated as early as age nine. Vaccination at earlier ages is beneficial by increasing earlier up-to-date status and providing immunity before HPV exposure (increasing vaccine efficacy). This study was successful in understanding parental perceptions of vaccinating children at 9–10 years of age. Many participants preferred starting the HPV vaccine series during their child’s teenage years; however, their rationale varied widely and was not scientifically supported. Prevalent themes from the qualitative analysis include concerns about vaccine side effects, limited knowledge and exposure to vaccine information, and mistrust in the healthcare system. Despite these opinions, many participants expressed trust in physicians who can provide an essential means of bridging the information and trust gap. We encourage emphasizing patient vaccine education through policy and public schooling, empowering physicians to educate patients on the HPV vaccine, and allowing PNs to play a vital role in patient education. Additionally, we propose further research into establishing school-based vaccination programs to increase the rate of HPV vaccination as well as other vaccines. Overall, this study provides insight into parental beliefs, attitudes, and understanding of HPV vaccination at a young age while suggesting potential interventions for improving HPV vaccine coverage.

## Figures and Tables

**Table 1 vaccines-12-00245-t001:** Demographics of patients whose parents completed and declined interviews from Texas HPV vaccine clinics.

	Declined N = 63	Interviewed N = 21	Total N = 84
Male	32 (51%)	10 (48%)	42 (50%)
Female	31 (49%)	11 (52%)	42 (50%)
Hispanic	20 (32%)	5 (24%)	25 (30%)
Non-Hispanic	43 (68%)	16 (76%)	59 (70%)
White	49 (78%)	10 (48%)	59 (70%)
Black	11 (17%)	10 (48%)	21 (25%)
Asian	1 (2%)	1 (5%)	2 (2%)
Other	2 (3%)	0 (0%)	2 (2%)
English	63 (100%)	21 (100%)	84 (100%)
9–10 years old	63 (100%)	21 (100%)	84 (100%)

**Table 2 vaccines-12-00245-t002:** Vaccine bias code specifications.

Code	Subcategories	Definition	Example Quote
Vaccines are safe and healthy		Individual mentions that vaccines are generally safe for the public or/and promote and achieve health by avoiding disease.	“I really think they are safe. My kids get their vaccines every time. Flu shot every year”.
Side-effect hesitancy		Individual mentions being weary or avoiding vaccinations because of possible side effects that these can have on the body.	“He catches something that he never had”.

**Table 3 vaccines-12-00245-t003:** Knowledge gap code specifications.

Code	Subcategories	Definition	Example Quote
Health Illiteracy		Individuals’ comments are either medically inaccurate or demonstrate a lack of understanding of the medical process.	“It [HPV vaccine] affects my ovaries…every month when I ovulate, I can tell which one is ovulating, and for about a week every month, my ovary feels like it’s on fire.”
Lack of HPV vaccine	Information	Individuals are unsure of what HPV is, including HPV prevalence, HPV vaccine research, and/or prevalence of disease.	“How long have we been doing this? How long have we been doing the research? How long has it been available if it’s available?”
	Exposure to information	Individuals have not been exposed to either positive or negative outside HPV vaccine sources of information.	“I honestly have not heard anything at all.”

**Table 4 vaccines-12-00245-t004:** Patient trust code specifications.

Code	Subcategories	Definition	Example Quote
Mistrust in news/social media		Individual mentions that social media or news are not a reliable source of information.	“There are a lot of stories that are false where people just go on and just like ‘oh the vaccine does this’ especially in regards to the COVID vaccines, which just simply aren’t true, and so not a good source to believe.”
Mistrust in healthcare system	Time	Individual shows a lack of trust within the healthcare system, in particular pertaining to the amount of time taken to develop medical innovations as a direct relationship with the effectiveness of the innovation.	“I want to hear we took our time, and we were able to pinpoint exactly what we needed and exactly what they needed. Time.”
	Risks	Individuals show a lack of trust in the healthcare system, in particular, in the possible risks that an individual might take when undergoing medical processes and the negative consequences that could arise.	“I mean you sometimes have to take the risk and I understood the risk when I got the COVID vaccine.”
	Experimentation	Individuals expressed the feeling that the medical industry uses people as a means to understand treatments through trial and error.	“She was like ‘dude, hello we all have to be a guinea pig sometimes.’ And I was like ‘what do you mean?’ She goes ‘well when the polio vaccine came out, do you think they were like, yeah this is going to work? No, there were hiccups here and there, they had to do their research. And now we have a really strong polio vaccine that can prevent that from happening.’”

**Table 5 vaccines-12-00245-t005:** Physician reliance code specifications.

Code	Subcategories	Definition	Example Quote
Trust in doctor	Information	Individual stated that they trust the information received regarding vaccination from their primary care physician.	“Oh, I was more than okay with it because I trust the doctor saying that hey this is going to help your child. So, when the lady came in and said can we do it, it would help her, I trusted her, and that made me do it. Anything the doctor says is good for my children, I’m all ears for.”
	Nurse vaccine administration	Individual stated that they would feel most comfortable with vaccines being administered at a physician’s office.	“No, because I know how bad the reactions can be. I would rather it happen at the doctor.”
Shared decision making		Individual states he/she feels more comfortable when medical professional incorporates patients into decisions regarding their care.	“That is the best way, kind of sharing the information ‘hey, this is what I saw, this is what we found.’ You can take the information, look it up yourself, be comfortable with yourself. Because I think that’s what we are missing, being comfortable with the decision that other people are taking for us, if that makes sense.”
General doctor administer vaccine		Individual expressed preference in having their child’s general doctor administer vaccinations compared to a specialty clinic	“I think that that really should be done by the main head doctor because, again, he’s the one that has been seeing the same kid for however many years, so they know that they know the child.”

**Table 6 vaccines-12-00245-t006:** Response to initiative code specifications.

Code	Subcategories	Definition	Example Quote
Trust in school nurse		Individuals are comfortable with nurses administering vaccines in schools without direct physician supervision.	“No, I can’t think of any [anything that would make me feel more comfortable about my child receiving the HPV vaccine at school]. A nurse is also a trained medical professional.”
Pharmacy HPV vaccination	Unwilling	Individuals expressed unwillingness to vaccinate their child against HPV at a pharmacy.	“No… I just don’t approve of it.”
	Not preferred	Individuals expressed hesitancy/concern to vaccinate their child against HPV at a pharmacy but would do it.	“I don’t typically love that because I just don’t see that as being super sanitary, but I mean, I would. It’s not something that I would not do, but I’d just prefer not to.”
Against mandatory vaccination		Belief that vaccinations should be up to the parent’s discretion and not be enforced by the government as a prerequisite for public benefits.	“No. Because some people are religious and they don’t believe in vaccines, and by y’all doing that, that’s stopping another way from not attending school for people so that they don’t need it, so that stops education for little ones in making the world dumb, honestly.”

**Table 7 vaccines-12-00245-t007:** HPV vaccination administration.

Code	Subcategories	Definition	Example Quote
Timing of HPV vaccine administration	Teens >10 years old	Individuals stated teenage years are the most appropriate age for HPV vaccination.	“Yes, 13, 14, you know when you are able to talk to them about sex, and you’re able to feel comfortable about things like that.”
	9–10 years old	Individuals are comfortable giving HPV vaccine between ages 9 and 10.	“I think it’s great. I think the reasoning that they give it to them earlier is a good reason.”
HPV vaccine for sexually active		Conception that individuals who would most benefit from vaccines are those who are sexually active or exploring their sexuality.	“I’m pretty sure I would probably recommend it to people who are active, sexually active, and then exploring their sexuality and whatnot.”

## Data Availability

Due to the qualitative nature of the data, we do not share it for confidentiality concerns.

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
