# Peer review of "Understanding Barriers to Human Papillomavirus Vaccination among Parents of 9–10-Year-Old Adolescents: A Qualitative Analysis"

_vaccines, 2024, doi:10.3390/vaccines12030245_

Round 1

Reviewer 1 Report

Comments and Suggestions for Authors

this manuscript describes results of 21 interviews about willingness to use HPV. My major concern is the very small sample size and consequently a low statistical significance of the conclusions. 

The authors did not describe how they choose the participants. Was it anybody that was available? Either way, this brings up questions about potential bias in the results.

In the interest of open science, I believe the complete set of questions as well as anonymized interviews should be made available. Currently, the authors present only selected quotes and there is no way for me as a reviewer to know that they did not select the quotes so that they support a priori hypothesis/conclusions. 

The paper misses a discussion where the authors describe how their results fit within the existing literature. While i am not an expert on HPV vaccine, there are numerous studies about reasons for vaccine hesitancy for all major kind of vaccines and I am sure HPV is no exception. The authors should compare their own results with the results of other authors. 

Reviewer 2 Report

Comments and Suggestions for Authors

The issues of vaccine reluctance have become acute during the COVID vaccination drive. The existing hesitance has grown, and the trust in vaccines has been a threat to public health. This is why qualitative research regarding parental attitudes is important.

The authors should include, either in the paper or as an appendix the guide for the interviewer, as it is not clear what questions were asked.

The 21 caretakers were selected by the authors, was there some bias? How many were approached? Could a table of the demographics of the subjects be added?

Reviewer 3 Report

Comments and Suggestions for Authors

This manuscript provides an insightful qualitative analysis into the parental barriers to HPV vaccination for 9–10-year-olds, a subject of critical importance given the low vaccination rates in the U.S. The study's methodology, employing semi-structured interviews to gather comprehensive data from parents, is both appropriate and robust, offering a nuanced understanding of the complexities surrounding HPV vaccine hesitancy.

The findings highlight important areas for intervention, such as the need for improved communication about the vaccine's benefits and safety, which can inform public health strategies to increase vaccination rates.

While the sample size is adequate for qualitative research, the diversity of the participant group could be better described. Information on the participants' demographic backgrounds could help understand how beliefs and attitudes may vary across different communities.

The study could benefit from a more detailed discussion on how its findings compare with existing research, particularly studies from different geographic or socio-economic contexts.

Recommendations for future research and potential public health interventions are valuable but could be expanded. Specifically, strategies to address the identified barriers more effectively could be further elaborated.

Overall, this manuscript makes a significant contribution to the field of public health and vaccination research. It not only sheds light on the reasons behind HPV vaccine hesitancy among parents of young adolescents but also offers a foundation for developing targeted interventions to improve vaccination rates. Further research in diverse settings and among broader populations could enhance the generalizability of these findings.

Round 2

Reviewer 1 Report

Comments and Suggestions for Authors

The authors addressed some of my earlier concerns. 

I still believe that at least the questions should be published as a supplementary material and a anonymized responses with identifying info blacked out should be made available too. I understand that authors cite IRB, but when I searched their submission, there is no IRB number or anything like that.
